# Targeting Sphingosine 1-Phosphate Metabolism as a Therapeutic Avenue for Prostate Cancer

**DOI:** 10.3390/cancers15102732

**Published:** 2023-05-12

**Authors:** Saida Mebarek, Najwa Skafi, Leyre Brizuela

**Affiliations:** 1CNRS UMR 5246, INSA Lyon, Institut de Chimie et Biochimie Moléculaires et Supramoléculaires (ICBMS), 69622 Lyon, France; saida.mebarek@univ-lyon1.fr; 2CNRS, LAGEPP UMR 5007, University of Lyon, Université Claude Bernard Lyon 1, 43 Bd 11 Novembre 1918, 69622 Villeurbanne, France; najwaskafi91@gmail.com

**Keywords:** sphingosine 1-phosphate, sphingosine kinase, prostate cancer, bone metastasis

## Abstract

**Simple Summary:**

Prostate cancer (PC) is the most common cancer in men over 65 and the fifth leading cause of death by cancer in men. Sphingosine 1-phosphate (S1P) is a bioactive sphingolipid that regulates cell proliferation, survival, differentiation, cell cycle, and apoptosis. S1P content and S1P-transforming enzymes activities are deregulated in several diseases including obesity, diabetes, neurological and cardiovascular disorders, and especially cancer. The implication of S1P metabolism in PC has been explored over the last 20 years in several in vitro and in vivo models of PC. Here we summarized (i) the most important findings about the role of S1P metabolism and its members in the oncogenesis of PC; and (ii) the most efficient molecules targeting S1P metabolism under preclinical and clinical development for curing PC.

**Abstract:**

Prostate cancer (PC) is the second most common cancer in men worldwide. More than 65% of men diagnosed with PC are above 65. Patients with localized PC show high long-term survival, however with the disease progression into a metastatic form, it becomes incurable, even after strong radio- and/or chemotherapy. Sphingosine 1-phosphate (S1P) is a bioactive lipid that participates in all the steps of oncogenesis including tumor cell proliferation, survival, migration, invasion, and metastatic spread. The S1P-producing enzymes sphingosine kinases 1 and 2 (SK1 and SK2), and the S1P degrading enzyme S1P lyase (SPL), have been shown to be highly implicated in the onset, development, and therapy resistance of PC during the last 20 years. In this review, the most important studies demonstrating the role of S1P and S1P metabolic partners in PC are discussed. The different in vitro, ex vivo, and in vivo models of PC that were used to demonstrate the implication of S1P metabolism are especially highlighted. Furthermore, the most efficient molecules targeting S1P metabolism that are under preclinical and clinical development for curing PC are summarized. Finally, the possibility of targeting S1P metabolism alone or combined with other therapies in the foreseeable future as an alternative option for PC patients is discussed. Research Strategy: PubMed from INSB was used for article research. First, key words “prostate & sphingosine” were used and 144 articles were found. We also realized other combinations of key words as “prostate cancer bone metastasis” and “prostate cancer treatment”. We used the most recent reviews to illustrate prostate cancer topic and sphingolipid metabolism overview topic.

## 1. Prostate Cancer Disease: An Overview

### 1.1. Epidemiology of Prostate Cancer

Prostate cancer (PC) is the second most commonly diagnosed solid tumor and the fifth leading cause of cancer-related death in men worldwide, with 268,490 estimated new cases in the year 2022. Over 60% of patients are diagnosed with PC after the age of 65, whereas only 2% of cases occur in people under the age of 50 [1,2]. Serum-prostate-specific antigen (PSA) is used as a diagnostic biomarker for patients with PC. However, it has been shown to lack specificity and therefore, lead to PC overdiagnosis and overtreatment [3]. The survival rate of PC patients is dependent on early disease diagnosis and available treatment options [4]. Most PC cases are identified at an early stage, with slow-growing tumors that are confined to the prostate. However, in approximately 20% of patients, tumors are not discovered before the metastatic state. Moreover, advanced PC cases often progress despite androgen ablation and are considered castration-resistant, evolving towards a metastatic phenotype, which will consequently lead to death [5,6,7].

### 1.2. The Cellular and Molecular Mechanisms in Prostate Cancer: Bone Metastasis 

Cancer metastasis is a process by which tumor cells detach from the primary tumor site and move to secondary sites via the blood circulation or lymphatic system to secondary sites where they colonize and form secondary tumors. Based on the “seed and soil” theory, which was first proposed by Stephen Paget in 1889, PC has a specific affinity to the bone. Indeed, bone is a fertile and productive soil due to the presence of growth factors and cytokines like transforming growth factor beta (TGF-β), bone morphogenetic proteins (BMPs), and interleukin-11 (IL-11) produced by osteoblasts. Those factors are conserved in the bone matrix, and released during physiological bone degradation, thus becoming free and accessible for cancer cells [8,9]. When PC spreads to the bone, it becomes incurable. PC-derived bone metastases are often characterized as osteoblastic but usually have an underlying osteoclastic component. They can induce adverse complications, called skeleton-related events (SRE), including pain, pathological fractures, and paralysis of the spine, which decreases the quality of life and worsens prognosis [10]. 

#### 1.2.1. Preparation of the Metastatic Niche

Before metastasis, a pre-metastasis niche is established in the target organ, providing a suitable microenvironment to support the colonization of metastatic tumor cells. Meanwhile, some tumor cells at the primary site undergo epithelial-to-mesenchymal transition (EMT) and release several extracellular vesicles (EVs), notably exosomes, that are involved in cell-to-cell communication [11]. During EMT, E-cadherin expression is downregulated in tumor cells, resulting in decreased adhesion and loss of apical polarity and basal anchoring. These events enable tumor cells to easily leave the primary lesion and migrate to other organs [12]. 

#### 1.2.2. Metastatic Homing

The first step of metastasis requires the escape of metastasizing tumor cells from the primary tumor into circulation. This invasion and intravasation are driven by tumor cell-intrinsic mechanisms, their associated stroma, and the local extracellular matrix (ECM) surrounding the primary tumor [13,14,15]. The suitable pre-metastatic niche must evolve to allow tumor cell engraftment (metastatic niche) and proliferation (micro- and macrometastatic transition). Fibroblasts, through fibronectin, attract hematopoietic cells expressing vascular endothelial growth factor receptor-1 (VEGFR-1) from the bone marrow and establish a metastasis-supporting microenvironment. Tumor cells accumulate on the endothelial surfaces within the circulatory system and subsequently undergo transendothelial migration, which is considered a key process in cancer metastasis. Once bound to the endothelium, cancer cells begin opening the endothelial junctions in response to multiple factors including TGF-β and VEGF, traversing the basement membrane and entering into the stroma [13,14,15]. The endothelial cells of the bone perivascular niche modulate cell trafficking. Once they arrive in the bone marrow, cancer cells require phenotypical changes to become stable and survive. They acquire the capability to respond to physical and chemical microenvironmental stimuli and adhere to the special niches previously prepared. 

#### 1.2.3. Bone Colonization 

The vicious cycle describes the communication between tumor cells and bone cells which results in the disturbance of bone homeostasis [16]. Tumor cells can acquire an osteoblast-like phenotype, also known as osteomimicry. To sum up, PC cancer cells that have migrated to the bone adapt to and modify, the surrounding microenvironment by secreting active molecules that promote osteoblast differentiation and proliferation, including TGF-β and endothelin-1. In turn, enhanced osteoblast activity drives tumor progression by releasing insulin-like growth factor (IGF-1), IL-6, and IL-8. Simultaneously, PC cells activate osteoclastogenesis via pro-osteolytic factors, such as receptor activators of nuclear factor kappa-Β ligand (RANKL). Accelerated bone resorption promotes the release of TGF-β and IGF-1, which further support cancer cell proliferation. Also, the release of osteolytic factors, such as parathyroid hormone-related peptide (PTHrP), activates osteoblast differentiation and proliferation, supporting bone formation even in the presence of a growing tumor [13,14,15]. 

### 1.3. Extracellular Vesicles: Crucial Messengers in Prostate Cancer–Osteoblast Crosstalk

Although multiple studies have focused mainly on the function of soluble factors in PC progression, new evidence has reported the critical role of EVs in tumor metastasis and bone colonization. EVs represent a heterogeneous population of membrane-derived vesicles released by essentially all cell types [17]. Their major function is the transportation of active molecules to recipient cells, thus influencing their physiological properties. Exosomes are defined as nanosized EVs, between 30 and 150 nm in diameter, that originate and are released from cells after multivesicular bodies (MVB) fuse with the plasma membrane. Exosomes consist of a lipid bilayer membrane enclosing a cargo of biomolecules, including proteins, lipids, RNA, and DNA [18]. Increasing evidence suggests that exosomes mediate intercellular communication by delivering their cargo to recipient cells to modulate target cell functions. Among all the biomolecules carried by the EVs, miRNAs have an important regulatory role in cancer progression and metastasis. 

The miRNA miR141-3p, released by the humanized PC cell line MDA-PCa-2b, promoted osteoblastic activity in vitro and created a favorable microenvironment for bone metastasis in vivo [19]. Since this study, various exosomal miRNAs derived from PC tumor cells have been implicated in the progression of bone metastasis [20]. Recent studies showed that PC-derived EVs facilitate tumor progression by instructing osteoblasts toward a protumorigenic cell type. In this way, EVs ensure the formation of an attractive and supportive niche for tumor growth and survival. Also, the presence of EVs in many body fluids could render them attractive biomarkers for the early detection of the disease. Thus, advances in this field will be useful to further deconstruct the complexity of the bone–PC microenvironment [21].

### 1.4. Summary of Treatments in Prostate Cancer

The recent advances in chemo-, immuno-, and hormone therapy have expanded the treatment options available for patients with advanced PC. PC treatment encounters many challenges due to different factors, including high genomic heterogeneity, immunosuppressive tumor microenvironment, and androgen-independence mechanisms [22,23,24]. Treatment plans may vary largely from one patient to another. Different options exist to reduce the progression and spread of cancer to the bones: chemotherapy combined with pain killer medications (like anti-inflammatory drugs or opioids), corticosteroids, external radiation, radiopharmaceuticals to reduce tumor size and pain, bisphosphonates, making bones stronger (ex: zoledronate), and anti-RANKL (denosumab) [25,26]. 

Surgery and radiotherapy are the standard treatments for PC. Several therapeutic options are available for pretreated patients with hormone therapy or chemotherapy (docetaxel): cabazitaxel, olaparib, and rucaparib for PC patients with BRCA mutations, Radium-223 for specific patients who have symptomatic bone metastasis, the first approved vaccine for PC patients sipuleucel-T, and ^177^LuPSMA-617 (radiopharmaceutical directed to prostate-specific membrane antigen-presenting cells). Currently, the potential role of triplet therapies encompassing androgen deprivation therapy (ADT), chemotherapy, and androgen receptor (AR) target agents (ARTAs), is gaining more attention, especially for metastatic hormone-sensitive PC [25]. A lot of therapies induce the development of resistance mechanisms in prostate tumor cells. Mechanistically, radiotherapy is commonly used for PC treatment because radiotherapy induces ROS generation and increases DNA damage to reduce PC progression [27]. In the past 20 years, therapeutic cancer vaccines targeting antigens have been extensively proposed as treatments for PC. The objective of using antitumor vaccines is to induce tumor-specific immune responses that can suppress tumor cells. The majority of these therapies have been stopped at phase I studies, although nearly all vaccines were safe and showed some proof of immunological response. Overall, five approaches GVAX, DCVAC/PCa, a multi-epitope peptide vaccine, sipuleucel-T, and PROSTVAC have been studied in randomized III trials. Among these vaccines, sipuleucel-T was approved by the FDA as a treatment for metastatic castration-resistant prostate cancer (mCRPC) [28]. Different therapies can induce resistance in prostate tumor cells. Thus, the discovery of new therapies for the treatment of mCRPC patients is necessary to improve medical care and nursing. 

## 2. Sphingosine 1-Phosphate Metabolism and Mode of Action

Sphingolipids are a large family of lipids, all of which are characterized by a long chain sphingosine base. Initially attributed with only a structural role in the buildup and flexibility of membranes and some roles in energy metabolism, several members are now known to have important roles in basic cellular functions contributing to health and diseases and were identified as bioactive sphingolipids. Sphingosine 1-phosphate (S1P) is regarded as the end product of sphingolipid metabolism and one of the most biologically active sphingolipids [29,30].

S1P was proven to control different cellular functions including proliferation, survival, and differentiation. Consequently, it was found to be involved in a wide array of developmental and homeostatic processes including vascular development and bone homeostasis [31,32]. The importance of this molecule at the developmental level is manifested by the death of mice with defective sphingolipid metabolism which turns off the production of S1P [33]. Also, its level was found altered in many pathological conditions including diabetes, obesity, neurological, cardiovascular, inflammatory, and autoimmune diseases along with cancer [34,35,36]. Thus, many studies were directed toward using S1P inhibitors in the treatment of different pathologies [37,38,39]. 

S1P is produced by the phosphorylation of sphingosine via a sphingosine kinase (Figure 1). Two isoforms of sphingosine kinases exist in mammals (SK1 and SK2) with differences in subcellular localization. Subsequently, S1P produced by each isoform may have a different role, according to the cellular compartment in which it is produced [39,40]. Interestingly, it has been reported that SK1 induces inflammation and maintains the metabolic shift known as the Warburg effect and cell survival, which further enables the acquisition of cancer hallmarks in affected cells while SK2 exhibits a non-overlapping function [41]. 

The bulk of bioactive S1P is extracellular and acts in an autocrine, paracrine, and even endocrine way. Extracellular S1P are mostly produced by SK1 which translocates to the cellular membrane upon activation where it will be near to its substrate, the sphingosine. After production, S1P can be released from cells by different transporters, including general ATP-dependent ABC transporters or by a specific transporter, Spinster 2 (Spns2), depending on the cell type [42,43,44] (Figure 1). Extracellular S1P can act directly on the cells producing it (autocrine mode of action) or on nearby cells (paracrine mode of action), but can also be complexed with carrier proteins and transported to other faraway targets, thus gaining an additional endocrine function. In the circulation, S1P is mostly found bound to albumin or high-density lipoprotein (HDL), where it is produced mainly by, but not restricted to, erythrocytes, platelets, and vascular endothelial cells [29,45,46].

At the target cell, extracellular S1P acts via binding to one of its 5 G-protein coupled receptors, Sphingosine 1-phosphate receptor 1-5 (S1PR1-5). These membrane receptors are coupled to different G-proteins depending on the type and state of cells. Mainly, S1P_1_ signals via G_i_, while S1P_2_ and S1P_3_ can signal via G_i_, G_q_ or G_12/13_, and S1P_4_ and S1P_5_ can signal via either G_i_ or G_12/13_ (Figure 1). Different G proteins then activate different signaling pathways affecting cell proliferation, survival, motility, and metabolism. The complexity of this system is behind the fact that S1P is involved in different cellular functions depending on cell type and differentiation state. This makes it challenging to predict the action of S1P on a certain cell type. S1PRs are ubiquitously expressed in almost all cells, which confers S1P a role, minor or major, in most physiological and pathological processes [47]. 

S1P can additionally act intracellularly. Intracellular S1P can be produced by either SK1 or SK2 in the cytoplasm and by SK2 in the nucleus or mitochondria, where SK2 resides [37]. Different intracellular targets were identified for S1P lately. For example, S1P was found to bind and activate β-site amyloid precursor protein (APP) cleaving enzyme-1 (BACE1) leading to an increase in amyloid-β peptide (Aβ) production which is involved in the pathogenesis of Alzheimer’s disease (AD) [48]. Cytoplasmic S1P also binds to the transcription factor peroxisome proliferator-activated receptor γ (PPARγ) and activates the transcription of its target genes [49]. Mitochondrial S1P, produced by SK2, was shown to bind to prohibitin 2 (PHB2), a mitochondrial protein that regulates its assembly and function [50]. Moreover, nuclear S1P can bind to human telomerase reverse transcriptase (hTERT) and prevent its ubiquitination and degradation [51]. Finally, S1P was also proven to be involved in the epigenetic regulation of gene expression by inhibiting the action of Histone Deacetylase HDAC1/2, which confers S1P’s additional important roles in highly differentiated cells [39,52] (Figure 1). 

S1P can be degraded via one of two pathways; the first one is reversible in which S1P is dephosphorylated back to sphingosine under the action of S1P phosphatases (SPP1 and SPP2) or some less specific lipid phosphate phosphatases (LPPs). Alternatively, S1P can be degraded irreversibly by S1P lyase (SPL) into ethanolamine phosphate and trans-2-hexadecenal [40]. As modulators of the S1P level, the expression of these enzymes is crucial in determining cell fate and function.

Thus, the response of a cell to S1P depends on many factors including the level of the enzymes producing and degrading it, the cellular compartment in which it is produced, the level of expression of its transporter, the presence of S1P-producing cell nearby, and the level and type of S1P receptors expressed on the target cells [53]. 

On the other hand, S1P is not the only bioactive sphingolipid. Sphingosine, in its turn, is produced by the hydrolysis of the variable acyl chain from ceramides by ceramidases. Ceramides are also bioactive sphingolipids, with roles almost opposite to that of S1P. This, in fact, is what gave rise to the notion of a “sphingolipid rheostat” which determines the fate of the cell. In general, ceramides have been linked to apoptosis and cell cycle arrest whereas S1P is a pro-survival and pro-proliferative molecule [54]. However, now the situation is more complicated after discovering that ceramides with different side chains can have opposite effects on cell survival and proliferation [37].

Understanding the role of S1P in different diseases may open the avenue for novel therapeutic approaches. As the level of S1P is an important predictor of cellular death and proliferation, investigating its role in cancer would be promising.

## 3. Role of Sphingosine 1-Phosphate Metabolism in Prostate Cancer

S1P metabolism members participate in several steps of PC tumorigenesis. Here are detailed the most important functions for S1P, SKs, SPL, and S1PR discovered in several studies with in vitro and in vivo models of PC. 

### 3.1. Role in Cell Death and Apoptosis

The first evidence of a role of a sphingolipid metabolite in apoptosis in PC cell models was published 25 years ago. Igarashi and colleagues demonstrated that sphingosine was able to provoke apoptosis in androgen-independent human prostatic carcinoma DU-145 cells through the downregulation of bcl-X_L_ with a mechanism independent of protein kinase C (PKC). Two directly derived metabolites, C2-ceramide or S1P failed to induce the same effect [55]. 

In the same year, Mc Donnell and colleagues showed that the LNCaP cell line (a human prostatic metastatic cell line obtained from a metastasis located in a clavicular lymph node) overexpressing bcl-2, was protected against C2-ceramide-mediated apoptosis [56]. Accumulation of endogenous ceramide-induced either by pharmacological inhibition of acid ceramidase with N-oleoylethanolamine or by serum deprivation caused apoptosis in androgen-independent human prostatic carcinoma cell line PC-3 [57]. Treatment with pro-apoptotic sphingolipids C6-ceramide and sphingosine strongly reduced viability in LNCaP and PC-3 cells. In this study, the overexpression of bcl-2 (a common feature in PC patients) in LNCaP and PC-3 cells did not protect them against loss of cell viability [58]. Pharmacological inhibition of SKs with non-selective SK inhibitor N,N-dimethylsphingosine (DMS), and with selective SK1 inhibitor SKI-II; induced cytotoxicity and caspase-dependent and independent-apoptosis in PC-3 cells. Again, overexpression of bcl-2 was not able to rescue PC-3 cells from apoptosis [58]. Blocking SK1 with small interfering RNA strategy (siRNA) or pharmacologically with SKI-II induced apoptosis in LNCaP and PC-3 cells [59]. Furthermore, treatment with selective SK1 inhibitor, B-5354 compound (B-5354c), obtained from a marine bacterial species [60], triggered caspase-dependent apoptosis in LNCaP and PC-3 cells. On the contrary, B-5354c failed to induce apoptosis in LNCaP/SK1 and PC-3/SK1 overexpressing cells by switching sphingolipid rheostat towards S1P [61]. 

Fingolimod, a general modulator of S1P metabolism, was proposed as an antitumoral agent nearly 25 years ago [62]. Fingolimod-induced caspase-3-dependent apoptosis in androgen-independent DU-145 cells [62]. Later, the authors also showed that fingolimod induced the cleavage of caspase-9 and 8 in the same PC cell model [63]. Years later, Cuvillier and colleagues found that fingolimod increased caspase-3/7 activity and therefore apoptosis in androgen-independent PC-3 cells [64]. Moreover, overexpression of SK1 protected PC-3 cells from fingolimod-induced apoptosis [64]. Other authors have also confirmed the impact of fingolimod as an inducer of apoptosis in PC cell models [65].

### 3.2. Role in Cell Viability and Proliferation

Several studies have confirmed the important role of SK/S1P signaling in cell viability and proliferation in PC cell models. The androgen-independent PC-3 cell line has stronger SK1 activity, gene, and protein expression compared to the LNCaP cell line [66]. Knockdown of SK1 expression by siRNA or pharmacological inhibition of SK activity with DMS induced an important and significant inhibition of PC-3 cell proliferation [66]. PC-3 and DU-145 cells sustain also higher genic expression of S1P_1_, but express equal levels of S1P_2_ and S1P_3_ receptors [66,67]. On the contrary, PC-3 cells express higher protein levels of S1P_1_, S1P_2,_ and S1P_3_ compared to DU-145 and LNCaP cells [67]. Exogenous S1P stimulated proliferation more efficiently in serum-deprived androgen-independent DU-145 cells than in PC-3 cells through the activation of protein kinases Akt and ERK signaling downstream S1P receptors. S1P treatment did not induce proliferation in LNCaP cells [67]. Moreover, inhibition of the S1PR signaling cascade with pertussis toxin significantly diminished PC-3 cell proliferation [66]. Interestingly, S1P_5_ was recently proven to be implicated in S1P-stimulated mitosis [68]. Moreover, this receptor was found to be necessary for S1P-induced autophagy in PC-3 cells [69,70].

PC is a convenient target for primary prevention care, since it develops slowly, before the appearance of symptoms, providing a relatively long period for therapeutic intervention. An extensive number of studies have established the anticancer effects of polyphenols such as resveratrol, epigallocatechin-3-gallate, curcumin, quercetin, genistein, and much more in PC [71,72,73]. Resveratrol and epigallocatechin-3-gallate or a mixture of polyphenols from grapevine extract (Vineatrol, 47.5% of resveratrol) or from green tea (polyphenol E, 62% of epigallocatechin-3-gallate) decreased proliferation and cell viability by blocking ERK/Phospholipase D/SK1 pathway in PC-3 and C4-2B cells (a castration-resistant PC cell line isolated from the bone metastasis of a mouse xenograft inoculated with C4-2 cells, a subline of LNCaP) [74]. When SK1-overexpressing PC-3 cells were heterotopically implanted in *nude* mice, they generated larger tumors and were poorly affected by polyphenols treatment. Furthermore, the effect of oral administration of epigallocatechin-3-gallate or polyphenol E was directly related to tumoral SK1 inhibition, reduction of primary tumor volume, and the occurrence and number of metastases in an orthotopic PC-3/GFP model of PC [74]. Taken together, these results demonstrate that SK1/S1P pathway is a target of dietary green tea and wine-derived polyphenols in PC.

A few years later, a study demonstrated that also SK2 was implicated in mitogenic responses in PC cell models. ABC294640, a selective, competitive SK2 inhibitor, decreased viability, clonogenicity, and proliferation of LNCaP and PC-3 cells independently of the intrinsic apoptotic pathway [75,76]. ABC294640 notably reduced AR and Myc expression, two oncogenes directing PC tumorigenesis in LNCaP, C4-2, and 22Rv1 cells (a human PC cell line obtained from a xenograft gradually spread in mice after castration-induced regression and relapse of the parental androgen-dependent CWR22 xenograft) [76]. Treatment of TRAMP-C2 adenocarcinoma cells (an epithelial cell line isolated from the prostate of an adult, male transgenic mouse naturally developing PC) with ABC294640 also decreased cell viability, induced DNA fragmentation, and increased lysosomal acidification [77]. Moreover, ABC294640 reduced S1P and increased dihydroceramide content in TRAMP-C2 cells [77]. In vivo, oral treatment of ABC294640 by gavage (50 mg/kg/day) of *nude* mice bearing LNCaP heterotopic tumors resulted in a significant and persistent reduction of tumor growth [76]. Furthermore, treatment of ABC294640 (50 mg/kg five days per week, intraperitoneally, i.p.) significantly decreased tumor volume TRAMP-C2 xenografts in syngeneic hosts [77].

The effect of a general blockade of S1P metabolism with fingolimod was also evaluated in PC in vitro and in vivo models. Several studies have demonstrated that fingolimod significantly reduced cell viability, cell proliferation, and clonogenicity of LNCaP, PC-3, and DU-145 cells [64,78,79,80,81]. Daily treatment of fingolimod (10 mg/kg, i.p.) of CWR22R *nude* mice xenografts significantly reduced tumor volume and circulating PSA levels in serum after 20 days of treatment [79]. CWR22R is an androgen-resistant recurrent tumor derived from CWR22 after castration of the host [82]. Proliferation was successfully inhibited in recovered tumors with less PCNA (proliferating cell nuclear antigen) and Ki-67 positive cells [79]. The tumors from mice had increased levels of cleaved caspase-3 and bcl-2, two apoptosis markers, determined by immunohistochemistry (IHC) and were positive for TUNEL (deoxynucleotidyl transferase dUTP nick end labeling) reaction [79]. Moreover, fingolimod also reduced tumor volume of PC-3 heterotopically implanted into *nude* mice after 15 days of treatment (5 mg/kg, subcutaneously, s.c.) [65]. 

As mentioned before, ADT and ARTAs are generally the first treatments that PC patients receive in the earlier stages of the disease. Short androgen deprivation in androgen-dependent LNCaP cells reduced cell proliferation concomitantly with a decrease in SK1 activity whereas the stable overexpression of SK1 protected these cells against androgen depletion effects on proliferation. Furthermore, dihydrotestosterone after binding to AR, was able to stimulate SK1 activity through PI3 kinase (PI3K)/Akt signaling [83]. In contrast, the selective SK inhibitor, SKI-II, significantly decreased AR expression in LNCaP cells by a p53-independent oxidative-stress mediated mechanism [84] Finally, castration of severe combined immunodeficiency (SCID) mice orthotopically implanted with LNCaP/GFP cells strongly decreased tumor growth and SK1 activity [83]. 

Patients suffering from mCRPC, develop resistance to ARTAs such as enzalutamide or abiraterone in 20 to 30% of the cases [26]. AR^+^ cell lines LNCaP, C4-2B, and 22Rv1 showed resistance to enzalutamide although it was greater in C4-2B and 22Rv1 cells compared to LNCaP cells [85]. Treatment with enzalutamide combined with specific SK1 inhibitor PF-543 or with selective SK2 inhibitor ABC294640 was more efficient in decreasing cell viability than SK inhibitors alone [85]. PC patients harboring ARTAs resistance had elevated ceramide circulating levels which were correlated with a worse prognosis of cancer disease [85]. Transcriptomic analysis of tumors from a cohort of mCRPC patients under ARTAs therapy (n = 108) revealed a correlation between shorter ARTA treatment and higher expression of at least 5 sphingolipid genes directly implicated in ceramide/S1P metabolism [85]. Authors proposed that tumors from mCRPC patients could metabolize circulating ceramide into S1P, thus increasing the local proliferation of PC tumor cells [85].

Interestingly, long-term removal of androgens in LNCaP and C4-2B cells induced a continuous increase in SK1 expression and activity that were correlated with neuroendocrine (NE) transdifferentiation of LNCaP and C4-2B cells [83]. LNCaP and C4-2B were transformed into neuron-like cells with progressively increased chromogranin A (CgA) protein expression, which is the most trustworthy marker for prostatic NE differentiation, along with neuron-specific enolase secretion (NSE) [83]. These results were confirmed by IHC. In a patient with advanced PC (Gleason 9, T4NxM1); under complete androgen blockade, with a low level of serum PSA and the presence of NSE in serum, SK1 was colocalized with CgA in prostatic samples [83]. 

Recently, these results were confirmed by Hsieh and colleagues [86]. SK1 was colocalized with synaptophysin, a NE marker, in PC tissue microarrays (TMA) (n = 44) [86]. Patient-derived tumor explants (PDE) treated with S1P showed increased protein expression of several neuronal progenitor transcription factors (NPTFs) (RN2, EZH2, FOXA2, SOX2). Conversely, pharmacological inhibition of SK1 significantly reduced protein expression of NPTFs and increased RE-1 silencing transcriptional factor (REST) repressor complex expression which is considered the master regulator of NE differentiation in PC [86]. Moreover, CRISPR/Cas9 gene knockout of SK1 also augmented REST expression in NE PC cell lines such as PC-3 [86]. Finally, patients with advanced PC tumors expressing high levels of SK1 combined with low levels of REST (n = 42) showed significantly reduced overall survival when compared with patients having tumors with low levels of SK1 and high levels of REST (n = 42) [86]. Collectively, these studies highlight the central role of the SK1/S1P axis in NE differentiation in PC.

### 3.3. Role in Hypoxia

Highly hypoxic human cancer cell models such as U87 glioma cells, clear cell renal cell carcinoma models (ccRCC) such as CAKI-1 and A498 and androgen-independent PC-3 cells have increased SK1 mRNA expression, SK1 and activity and, consequently, increased S1P secretion under hypoxic conditions [87,88,89]. In PC-3 cells, SK1 activity was strongly and transiently increased under hypoxia (1% O_2_) [87]. Pharmacological inhibition of SK1 activity with SKI-II inhibitor or SK1 knockdown with siRNA strongly reduced hypoxia-inducible factor 1α (HIF-1α) expression and transcriptional activity in PC-3 cells [87]. Moreover, SK1 was found to regulate HIF-1α stability by activating Akt/glycogen synthase kinase-3β (GSK3β) signaling, and HIF-1α degradation through pVHL (von Hippel-Lindau tumor suppressor protein) dependent-proteasome pathway [87]. 

After this first study, other authors confirmed the pivotal role of SK1 activity in the regulation of HIF-1α levels under hypoxia. Melatonin hormone [90]; cumestrol, (a natural compound derived from soybean, brussels sprouts, or spinach with estrogenic properties) [91], pristimerin (a molecule found in Reissantia buchananii and other organisms) [92] and an ethanol extract obtained from Crataegus pinnatifida fruit mainly composed of chlorogenic acid [93]; were all able to inhibit SK1 activity and to suppress HIF-1α expression and the consequent Akt/GSK3β phosphorylation in PC-3 and/or DU-145 cells.

Blocking extracellular S1P either with an anti-S1P antibody (Sphingomab) or by reducing S1P release from cells with siRNAs directed against S1P-specific transporter Spns2; provoked a significant reduction in HIF-1α protein expression in hypoxic cells lines such as PC-3, human lung cancer cell line A549 and U87 glioma cell line [94]. In vivo, the treatment every other day for up to 9 days with Sphingomab (50 mg/Kg) significantly reduced HIF-1α and glucose GLUT-1 transporter expression in PC-3/GFP tumors from orthotopically transplanted *nude* mice compared to IgG-treated control mice [94]. Anti-S1P antibody was efficient after 5 days of treatment for triggering an “oxygenation window” where tumor surrounding vessels were somehow “fixed” and became less tortuous and leaky. Tumor blood flow and tumor-surrounding vasculature functionality were significantly augmented [94]. Moreover, initial treatment with Sphingomab for 5 days ameliorated the docetaxel effect, the standard chemotherapy given to advanced hormone-resistant PC patients, in PC-3/GFP orthotopically transplanted mice. Tumor volume, peri-aortic lymph nodes volume, and metastasis dissemination were significantly reduced in mice when Sphingomab was administrated before docetaxel compared to mice that received only docetaxel, only anti-S1P or the two treatments simultaneously [94]. 

### 3.4. Role in Radiosensitivity

Spiegel and colleagues were pioneers in exploring the role of different sphingolipid metabolites in the radiosensitivity of PC cells. LNCaP cells, which are radiation-insensitive, failed to have an increase in sphingosine and ceramide levels after 8-Gy irradiation and consequently, they did not undergo apoptosis. Exogenous sphingosine or SK inhibitor DMS was able to induce apoptosis and to sensitize LNCaP cells to gamma-irradiation-induced apoptosis [95]. Treatment with C2-ceramide or bacterial sphingomyelinase enhanced the effects of endogenous ceramide on TNF-α-induced apoptosis with or without gamma-irradiation [96]. Inhibition of SK1 activity by fingolimod-sensitized androgen-independent PC-3 and DU-145 cells to gamma-irradiation more efficiently than the selective SK1 inhibitor SKI-II [64]. Moreover, daily treatment with fingolimod (2.5 mg/kg, 14 days, i.p) radiosensitizer PC-3/GFP tumors orthotopically implanted in *nude* mice [64]. Tumors were significantly smaller after gamma-irradiation (4 × 5 Gy) and the number of metastasis per animal was reduced when animals were previously treated with fingolimod [64].

Years later, S1P cell content in PC cell models was modulated using a different strategy. Downregulation of SPL (the enzyme that degrades S1P irreversibly), by siRNA enhanced survival after gamma-irradiation by reducing the level of DNA damage sensing and repairing proteins. Conversely, overexpression of SPL sensitized C4-2B and PC-3 cells to gamma-irradiation [97]. 

### 3.5. Role in Chemosensitivity

Cuvillier and colleagues have extensively worked on the role of the SK1/S1P axis in different PC cell and animal models, especially related to chemotherapy resistance. In 2005, they demonstrated that camptothecin (a topoisomerase-I inhibitor) and especially the taxoid docetaxel (an inhibitor of microtubules depolymerization), provoked strong inhibition of SK1 activity, increased ceramide/S1P ratio and apoptosis in LNCaP cells and androgen-independent PC-3 and DU-145 cells [59,98]. Of note, another taxoid, paclitaxel, reduced PC-3 cell number after treatment and increased ceramide content, but without modifying S1P content [99]. Yet again, docetaxel was more efficient than camptothecin in inhibiting SK1 activity and increasing ceramide/S1P content in a murine orthotopic model of PC-3/GFP cells [59]. In contrast, overexpression of SK1 in LNCaP and PC-3 cells induced chemoresistance manifested by a reduction of apoptosis after chemotherapy in in vitro and in vivo models [59]. Moreover, docetaxel treatment decreased tumor volume and metastasis occurrence more efficiently than camptothecin. Authors concluded that the severity of SK1 inhibition could prognosticate the consequences of chemotherapy in PC. A moderate inhibition of SK1 activity indicated lower drug efficacy whereas a strong inhibition of SK1 activity was correlated with significant antitumoral efficacy [59].

A few years later, it was demonstrated that the selective inhibition of SK1 activity by B-5354c sensitized LNCaP and PC-3 cells to camptothecin-induced and docetaxel-induced apoptosis, respectively [61,98]. Furthermore, camptothecin or B-5354c treatment poorly reduced tumor size and metastasis occurrence in orthotopic PC-3/GFP in vivo model established in *nude* mice, whereas the combination of the two treatments had a synergic effect on tumor volume and metastasis spreading [61]. 

Increasing S1P content in PC cell models by blocking SPL expression, also induced resistance to chemotherapy. On the contrary, overexpression of SPL in C4-2B and PC-3 cells increased apoptosis after docetaxel treatment [97]. In summary, accumulation of S1P in prostate cancer cell models either by overexpression of SK1 or by SPL blockade was directly correlated with chemoresistance. Taken together, these studies consolidated the ceramide/S1P ratio as a “chemotherapy sensor” for predicting PC cell response after chemotherapy. 

### 3.6. Role in Metastasis

A few articles have explored the role of S1P metabolism in the onset and development of PC-derived metastasis. Treatment with S1P metabolic modulator fingolimod abolished the migration and invasion capacity and reduced stress fiber formation in androgen-independent PC-3 and DU-145 cells by a mechanism involving RhoA-GTPase [81]. 

Ogretmen and colleagues demonstrated that when TRAMP mice (transgenic adenocarcinoma of the mouse prostate model) are crossed with SK1 knockout (KO) mice, an inhibition in tumor growth and a decrease in the progression of the disease occur [100]. Tail vein injection of MB49 murine bladder tumor cells produced less lung colonization and metastasis when injected in SK1 KO mice [100]. Similar results were obtained after tail vein injection of murine B16 melanoma cells [100]. Partial blockade of SK1 expression (−60%) with siRNA strategy in MB49 cells before injection in mice flanks produced smaller tumors in wild type (WT) or in SK1 KO mice. In contrast, intravenously injection of MB49 cells after SK1 knockdown in WT or SK1 KO mice generated similar colonization and metastasis of the lung. Moreover, neutralization of systemic S1P with Sphingomab (7.5 mg/kg/day for 16 days) in the MB49 dissemination model significantly reduced lung metastasis (−60%) [100]. Therefore, this study implicated both systemic and tumor-derived S1P in local tumor development whereas systemic S1P was necessary for lung metastasis [100]. 

Bone is the prevalent site for metastasis in PC [14,101]. In 2009, the nuclear expression of the osteogenic marker Runx2 (runt-related transcription factor 2) was discovered to be up-regulated in PC [102]. Moreover, Runx2 expression was significantly increased in metastatic PC patients with a high Gleason score (≥7) [102]. Treatment with fingolimod significantly decreased Runx2 expression and cadherin switching reversal (increase in E-cadherin expression and decrease in N-cadherin expression) in androgen-independent PC-3 cells [102].

In 2014, the crucial role of the SK/S1P metabolic axis in PC-derived bone metastasis was demonstrated [103]. The authors found that murine pre-osteoblastic cell line MC3T3-E1 and mouse or human primary pre-osteoblastic cells exhibited strong SK1 activity. Moreover, strong SK1 expression was detected in human osteolytic bone [103]. S1P-containing conditioned medium from osteoblastic cells stimulated the proliferation of several PC cell models (LNCaP, C4-2B, 22Rv1, PC-3, and DU-145) through the S1P_1_ receptor. In addition, osteoblasts-derived conditioned medium increased the chemo- or radioresistance of PC cells. When the S1P content of conditioned medium from osteoblastic cells was reduced by different strategies (pharmacological inhibition of SK1 activity with SKI-II, siRNA strategy directed against SK1 or Spns2, S1P blockade with Sphingomab), it lost its ability to stimulate PC cell proliferation and to protect against therapies [103]. 

Interestingly, Gram-negative derived lipopolysaccharide (LPS), which induces inflammation in host cells, was able to activate and translocate SK1 by increasing phosphorylation of residue Ser225 of LNCaP, PC-3, and DU-145 cells. LPS also stimulated cell migration and invasion of PC cells by increasing serine protease activities, specially matriptase activity [104]. Mechanistically, SK1-derived S1P stimulated S1P_4_ and activated matriptase inducing cell invasion and metastasis in PC cell models [104]. Of note, chronic inflammation of the prostate (prostatitis), which can be caused by a bacterial infection, is frequently associated with pre-tumoral lesions in PC [105].

### 3.7. Role as Biomarker of Disease

Sphingolipid metabolism, especially sphingolipid rheostat (ceramide/sphingosine/S1P ratio), regulating enzymes (mostly SK1/SK2) and/or S1P_1-5_ receptors; have been proposed as diagnostic, prognostic, and/or severity biomarkers in several cancers (breast, colon, pancreas, liver, glioma, etc.). The first study where the SK1/S1P metabolic pathway was proposed as a new biomarker of PC disease appeared in 2010. A clinical study realized with 30 prostatectomy specimens revealed a significantly higher SK1 expression and activity compared with normal prostate epithelium samples. Semi-quantitative analysis by IHC showed that SK1 expression was correlated with tumor grade determined by Gleason score. Moreover, treatment failure as defined by adjuvant radiation or ADT or biochemical recurrence was correlated with higher SK1 activity [106]. Recently, McGowan and colleagues pointed out that there is a difference of expression between the two major expressed forms of SK1, SK1a and SK1b isoforms (with a supplementary 86 aa sequence at N-ter) [107]. While SK1a is thoroughly expressed in several cancer cell tissues, only SK1b is present in tissues from breast cancer, lung cancer, or PC [107]. Of note, tumoral SK1 activity was implicated in a bone metastatic spread in these cancer models [101].

Two years later, in 2012, the same team demonstrated a significant decrease in protein expression and activity of S1P-irreversible degrading enzyme SPL in PC tumor samples compared with normal adjacent tissues. SPL protein expression and activity were also directly correlated with the Gleason score after tissue microarray (TMA) analysis and inversely with SK1 protein expression activity [97]. Consequently, PC patients with advanced disease featured the lowest SPL and the highest SK1 activity suggesting an important and significant accumulation of S1P in tumor tissues and surroundings [97]. Moreover, SPL and/or SK1 expressions were independently predictive of the aggressivity of PC, after analysis of TMA, reinforcing the theory of S1P putative role in PC oncogenesis.

Intriguingly, biochemical analysis of S1P content in plasma from a cohort of PC patients (CONSORT, n = 88) revealed significantly lower levels of plasma S1P compared to age-matched benign prostate hyperplasia (BPH) patients (n = 110) or healthy volunteers with no cancer history (n = 20) [108]. S1P plasma content was inversely correlated with the TNM stage of the disease, with positive lymph node and PSA content. Contrariwise, S1P plasma content was neither correlated with Gleason score nor with metastasis occurrence [108]. S1P plasma levels were reduced in PC patients with advanced disease dismissing the prospect of circulating S1P to be derived from cancer cells. RBCs are known as the major source of S1P in blood [109,110]. S1P decrease in plasma of PC patients was correlated with a decrease in SK1 activity in erythrocytes, probably as a consequence of cancer development and anemia, which frequently accompanies PC disease. Authors suggested S1P plasma level from PC patients as an earlier diagnostic and prognostic marker compared to PSA [108]. 

Metabolomic and transcriptomic analysis of prostatic tissues from PC patients (n = 55) compared to prostatic tissues from BPH patients (n = 16) demonstrated a high increase in sphingosine content [111]. Furthermore, sphingosine was highly suggested as a biomarker for distinguishing PC patients from those suffering from BPH, especially in patients with low levels of PSA (0–10 ng/mL) [111]. S1PR_2_ expression, which is considered a tumor suppressor because of its inhibitory effect on cell migration and metastatic dissemination through blockage of Rac signaling or activation of the Rho pathway, was downregulated in prostatic tissues from PC patients (n = 25) compared to their adjacent noncancerous tissues [111].

Conversely, a recent study realized in a large cohort of PC patients (n = 146) with a 13-year follow-up of the disease, found a decrease in plasma sphingosine and sphinganine (dihydrosphingosine) compared to matched controls (n = 272). The level of sphingolipid content in plasma was performed with ultra-high-performance liquid chromatography-high resolution mass spectrometry (UHPLC-HRMS) [112]. Additionally, similar results were obtained by liquid chromatography coupled with tandem mass spectrometry (LC-MS/MS)-based non-targeted metabolomics analysis in serum from PC patients (n = 72) compared to BPH patients (n = 72) [113]. Sphingosine and sphinganine levels were significantly reduced in patients with low levels of PSA (4–10 ng/mL) [113].

## 4. Targeting SK/S1P Metabolism as a Therapeutic Option for Prostate Cancer 

### 4.1. N,N-Dimethylsphingosine (DMS) 

The first molecule used to target SK activity in PC models was DMS (N,N-dimethylsphingosine) [58,95], a nonspecific pharmacological inhibitor of SKs (Figure 2). DMS can inhibit SK activity in several cancer cell models (K_i_ = 2.3–6.8 µM) [114,115] and it is selective over PKC when used at less than 10 µM in PC12 cells (a cell line derived from a pheochromocytoma of the rat adrenal medulla) [114]. DMS inhibited SK activity, increased sphingosine content, and sensitized PC cell models [58,95] and a wide array of cancer cell lines [116] to apoptosis. DMS has not been used in PC in vivo models but it was able to reduce tumor growth in mice s.c. inoculated with human gastric carcinoma cell line MKN74 [117] or with human epidermoid carcinoma KB-3-1 [118]. 

### 4.2. Safingol

Another non-selective SK inhibitor, safingol (L-threo-dihydrosphingosine or [(2S,3S)-2-amino-1,3-octadecanediol]) (IC_50_ = 5 µM for SK1) [119] (Figure 2), which is also a specific inhibitor of PKCβ-I, PKCδ, PKCε and PI3K activity [30] was tested in PC models. Safingol was cytotoxic and anti-proliferative in several PC cell lines [120,121]. TRAMP mice fed with safingol presented important liver and renal toxicity without any reduction in prostate pre-neoplastic lesions (prostate intraepithelial neoplasia, PIN) [122]. The combination of safingol with traditional chemotherapy agents mitomycin-C [123], camptothecin [124], or doxorubicin [125] enhanced their effects by inducing apoptosis and ROS production in different cancer cells. Two phase I trials have been performed with safingol, one in combination with doxorubicin [126] and another with cisplatin [127]. Both clinical trials resulted in slight improvements in patients’ status. Similarly to pre-clinical trials, hepatotoxicity was detected [127]. Safingol was proposed to be a therapeutic option when combined with chemotherapy [128]. Resistance to oxaliplatin, cisplatin, and docetaxel was related to an increase in SK1 activity and a decrease in SPL activity resulting in an overall increase in S1P content in different gastroesophageal cancer cell lines [128]. This phenomenon has been already observed in PC cell models and tumor samples from PC patients [59,61,98,103]. In conclusion, safingol could help to overcome cytotoxic drug resistance in chemoresistant cancers such as gastroesophageal cancer or PC. 

### 4.3. Enigmol

Enigmol ((2S,3S,5S)-2-amino-3,5-dihydroxyoctadecane) (Figure 2), is a synthetic, orally active sphingosine analogue. Instead of being phosphorylated by SKs, enigmol is slowly N-acetylated by ceramide synthase [129]. This condition confers to enigmol stability and retards its degradation [129]. Enigmol significantly reduced cell viability and increased cell toxicity in LNCaP, PC-3, and DU-145 cells [130,131]. Moreover, its antiproliferative and cytotoxic activities were credited to the inhibition of SKs [131], PKCs [132], and ceramide synthase [129]. Enigmol, when administered i.p., suppressed tumor growth in *nude* mouse xenograft models for PC-3 or DU-145 cells [131]. Furthermore, *nude* mice bearing s.c. tumors of LNCaP or PC-3 cells also reduced tumor volume after receiving enigmol or its analogs daily by oral gavage [130]. Enigmol and its derivates did not affect mice’s weight and were as efficient as ADT or docetaxel in inhibiting prostatic tumor growth [130]. Finally, a fluorinated enigmol (CF2-enigmol), accumulated more than enigmol in heterotopic PC-3 tumors from *nude* mice without systemic toxicity [133]. Therefore, CF2-enigmol significantly reduced tumor growth better than enigmol [133]. As enigmol was equally effective as docetaxel in reducing prostatic tumor volume [130]; CF2-enigmol could have stronger effects than docetaxel in PC [133]. CF2-enigmol was also highly concentrated in the brain, lungs, and kidneys, thus opening the possibility of an antitumor effect of CF2-enigmol in solid tumors located in these organs [133].

### 4.4. SKI-II

In the early 2000s, new SK inhibitors (SKI) were discovered by French and colleagues [134]. Five compounds of SKI (I–V) have been synthesized so far [134]. All of the SKI compounds are able to inhibit SK activity and to induce apoptosis in cancer cells [134]. SKI-I, SKI-II, and SKI-V had significant anti-tumor activity notably in mice models for breast cancer or cervical cancer [135,136]. SKI-II, also called SKI ((2-(p-hydroxyanilino)-4-(p-chlorophenyl)thiazole)) (Figure 2), is the most studied SK inhibitor alone or combined with another chemotherapeutic agent [30]. SKI-II is an orally available, dual SK1/SK2 inhibitor, capable of inhibiting human recombinant SK1 activity non-ATP-competitively (IC_50_ = 0.5 µM) [134]. In cells, SKI-II inhibits both SK activities with similar efficacy (K_i_ of 16 µM and 8 µM for SK1 and SK2, respectively) [134]. SKI-II was found to be selective for SK1 over other lipid kinases (SK2, PI3K, ERK2, PKCα) at concentrations up to 60 µM in different in vitro and in cellular assays. SKI-II induced proteasomal degradation of SK1 two isoforms (SK1a and SK1b) in androgen-sensitive LNCaP-AS cells [137]. SKI-II-mediated proteasomal degradation of SK1 was dependent on ceramide and induced apoptosis of LNCaP-AS cells [137]. SKI-II was less efficient in inducing SK1b proteasomal degradation and related apoptosis in androgen-independent LNCaP-AI cells but it increased p53 and p21 expression and induced growth arrest in the same model [137,138]. Furthermore, SKI-II showed different cytotoxic, antiproliferative, and apoptotic effects in PC cellular and in vivo models [58,59,61,87,98]. For instance, there are no clinical trials testing SKI-II in cancer.

### 4.5. ABC294640

ABC294640 (Opaganib, Yeliva^®^, (3-(4-chlorophenyl)-N-(4-pyridinylmethyl)-tricyclo [3.3.1.1^3,7^]decane-1-carboxamide) (Figure 2), is an orally available first-in-class selective SK2 inhibitor (K_i_ = 9.8 µM) with antitumor activity, competitive with respect to sphingosine [139]. ABC294640 inhibits SK2 activity (IC_50_ = 60 µM) without affecting SK1 activity up to 100 µM and it has no effect on 50 different lipid kinases up to 50 µM [139]. ABC294640 acts as a mimetic of sphingosine being able to inhibit simultaneously SK2 and dihydroceramide desaturase (DES1) [77]. Several promising in vitro and preclinical studies have demonstrated the antitumoral activity of ABC294640 [30]. ABC294640 induced autophagy in A498 and PC-3 cells resulting in non-apoptotic cell death and significantly reduced tumor growth in female *nude* mice injected s.c. with A498 cells [140]. Moreover, ABC294640 strongly decreased the proliferation of PC in vitro and in vivo models in a dose-dependent manner [75,76]. 

Several clinical trials have been performed to evaluate the impact of AB294640 combined with classic chemotherapy or alone in tumorigenesis of different cancers. In the first non-targeted phase I trial [141], 21 patients with advanced solid tumors (3 of them with urothelial cancers) received escalating doses of ABC294640 for 28 days (250 mg, 500 mg, or 750 mg twice a day). The most common symptoms were fatigue, nausea, vomiting, and diarrhea. Lipidomic analysis of plasma showed that ABC294640 reduced S1P circulating levels during the first 12 h returning to baseline levels after 24 h [141]. 15 of 21 patients completed at least 2 cycles of treatment and were evaluated for efficacy. 6 patients experienced stabilization of the disease during 6 months but 8 patients had progressive disease [141]. Recently, another phase I trial in relapsed and/or refractory multiple myeloma (RRMM) was completed [142]. 13 patients with RRMM who received immunomodulatory drugs and proteasome inhibitors as therapy were enrolled. 3 doses were again tested as in the previous phase I trial [141]. No related adverse events such as a decrease in circulating neutrophiles, were detected. One patient has a very promising partial response and another patient showed stable disease for 3 months [142]. Interestingly, there is an ongoing phase II trial in mCRPC combining ABC294640 with 2 AR antagonists, enzalutamide and abiraterone (for more details, check clinicaltrials.org, NCT04207255).

### 4.6. Fingolimod 

Fingolimod (FTY720, Gilenya^®^, 2-amino-2-[2-(4-octylphenyl) ethyl]propane-1,3-diol) (Figure 2), is an oral immunosuppressor drug approved in 2010 by the FDA for the treatment of patients suffering from relapsing/remitting multiple sclerosis [143,144]. In 2018, the FDA expanded the approval of fingolimod to treat multiple sclerosis in pediatric patients [145]. Fingolimod is a structural analog of sphingosine and can be phosphorylated in vivo, principally by SK2 [146] to form FTY720-phosphate, a mimetic of S1P. FTY720-phosphate interacts with all S1P receptors except S1P_2._ Its action is principally mediated through S1P_1_, inducing its downregulation and proteasomal degradation [147]. Downregulation of S1P_1_ sequestrates lymphocytes in secondary lymphoid organs or thymus provoking general lymphopenia and thus immunosuppression [148,149,150]. Fingolimod, under its unphosphorylated form, inhibited SK1 activity (IC_50_ = 5–12.5 µM) in different cell models including PC cell models [64,151,152] by a mechanism implicating its proteasomal degradation [152]. Moreover, fingolimod inhibited SK2 activity in several cell types such as osteoblasts, chondrocytes, or macrophages (IC_50_ = 2.5–10 µM) [151,153]. Fingolimod is therefore a functional antagonist of S1P_1_ and is considered a general modulator of S1P metabolism.

Daily treatment with fingolimod (1 mg/kg, orally) suppressed prostatitis in a rat experimental autoimmune model after 15 days of treatment [154]. Tumor growth was significantly reduced in CWR22R or PC-3 *nude* mice xenografts after i.p. or s.c. treatment with fingolimod [65,79]. Treatment with encapsulated fingolimod combined with chemotherapeutic drug docetaxel in lipid–polymer nanoparticles was significantly more efficient than docetaxel or fingolimod alone in reducing tumor volume of PC-3 cells subcutaneously injected in SCID gamma (NSG) immunodeficient male [155]. Interestingly, when fingolimod was encapsulated, it was unable to induce lymphopenia in mice [155]. Furthermore, fingolimod has been recently included as a theragnostic small-molecule prodrug conjugate, which can be eventually used as a therapy for mCRPC patients [156].

Since its validation by the FDA for the treatment of multiple sclerosis, fingolimod has been used in various clinical trials, not only for the treatment of autoimmune diseases such as multiple sclerosis but also in cancer. Fingolimod safety in combination with radiation and temozolomide was evaluated in a phase I clinical trial for newly diagnosed patients with high-grade glioma (NCT02490930). Moreover, there is an ongoing phase I trial to analyze the effect of fingolimod in breast cancer patients receiving paclitaxel (NCT03941743). In this case, fingolimod is used because of its immunomodulatory effects, notably to prevent neuropathic pain caused by paclitaxel. There are not currently clinical trials involving fingolimod and PC. 

### 4.7. Sphingomab/Sonepcizumab

Anti-S1P antibody Sphingomab was first proposed as a potential antitumor therapy in an elegant study published in 2006 by Sabbadini and colleagues [157]. Authors demonstrated a significant reduction in tumor growth after treatment with anti-S1P antibody (25 mg/Kg, every 3 days, i.p.) for 28 days in orthotopic models of breast cancer (MDA MB-231, MDA-MB-468), ovarian cancer (SKOV3) and in an s.c. model of lung adenocarcinoma (A-549) [157]. Sphingomab was as efficient as weekly treatment with taxol derivates in diminishing tumor volume [157]. Specifically, Sphingomab targeted endothelial cells from the tumor microenvironment inhibiting the release of proangiogenic factors (IL-6, IL-8, VEGF) and therefore reducing tumor vascularization [157]. Ten years later, anti-S1P antibody showed its efficacity in reducing tumor growth in RCC xenograft mice [158] and as shown before, in PC xenograft mice [94].

Using an anti-S1P antibody as a new therapy for treating cancer patients seemed promising. First, it reduced the bioavailability of S1P at its receptors from tumor cells resulting in an inhibition of tumor proliferation. Second, it decreased tumor angiogenesis by inhibiting the secretion of angiogenic factors from endothelial cells. The humanized version of the anti-S1P antibody Sphingomab, Sonepcizumab, showed excellent characteristics with respect to thermal stability, kinetic and binding specificity [159]. Moreover, Sonepcizumab was evaluated in a competitive ELISA for cross-reactivity against over 60 species including bioactive lipids and other molecules of interest [160]. Sonepcizumab was highly specific to S1P; nevertheless, it also recognized a reduced form of S1P (dihydroS1P also called sphinganine) and phosphate ester group added to the sphingoid base forming sphingosylphosphoryl choline [160].

Sonepcizumab has been the object of five clinical trials (for more information check clinicaltrials.org) including a phase 2 study where Sonepcizumab was proposed as monotherapy in patients (n = 40) with advanced, unresectable, refractory RCC who have previously failed up to three therapies, including VEGF and/or mammalian target of rapamycin (mTOR) inhibitors. Even if the major point of the study was not reached, 2-month disease free and an overall survival of 21.7 months were registered [161].

Unfortunately, Lpath company stopped the preclinical development of Sonepcizumab in 2017.

## 5. Conclusions and Future Directions

Targeting S1P metabolism in cancer especially in PC seems promising. SK1 or SK2 specific inhibitors and S1P modulators, alone or combined with ARTAs (abiraterone, enzalutamide), chemotherapeutic agents (cabazitaxel), vaccines (sipuleucel-T) or radiopharmaceuticals (^177^LuPSMA-617) could be a therapeutic option for mCRPC patients. Currently, a phase II trial in mCRPC patients combining ABC294640 selective SK2 inhibitor with 2 AR antagonists, enzalutamide and abiraterone is active (NCT04207255). On the other hand, there are no ongoing clinical trials in cancer implicating SK1-specific inhibitors such as PF-543. Interestingly, the anti-tumoral effect of dual inhibitors of SK1/SK2 activity [162] are now being investigated when SK1 and SK2 are both important therapeutic targets as in non-small cell lung cancer (NSCLC) [163]. Fingolimod could be the next S1P modulator to be tested in mCRPC patients. There is an ongoing phase I trial to analyze the effect of fingolimod in breast cancer patients receiving paclitaxel (NCT03941743). Breast cancer and PC are hormonal cancer types with significant similarities (development of hormone resistance, metastasis to the bone). Even if the main objective of this clinical trial is to analyze the effect of fingolimod as an immunomodulator in paclitaxel-related neuropathic pain; its impact in disease overall evolution cannot be excluded. Finally, other S1P metabolism modulators such as siponimod (functional antagonist of S1P_1_ and S1P_5_), ozanimod (a potent agonist of S1P_1_ and S1P_5_), ponesimod (functional antagonist of S1P_1_), ceralifimod (selective S1P_1_ and S1P_5_ modulator), GSK2018682 (selective S1P_1_ and S1P_5_ modulator) and amiselimod (MT-1303) (selective S1P_1_ modulator), deserve to be explored in cancer, especially in PC [148]. Of note, S1P_5_ could be an interesting target in PC because of its implication in S1P-stimulated mitosis [68] and S1P-stimulated autophagy of androgen-independent PC-3 cells [69,70].

## Figures and Tables

**Figure 1 cancers-15-02732-f001:**
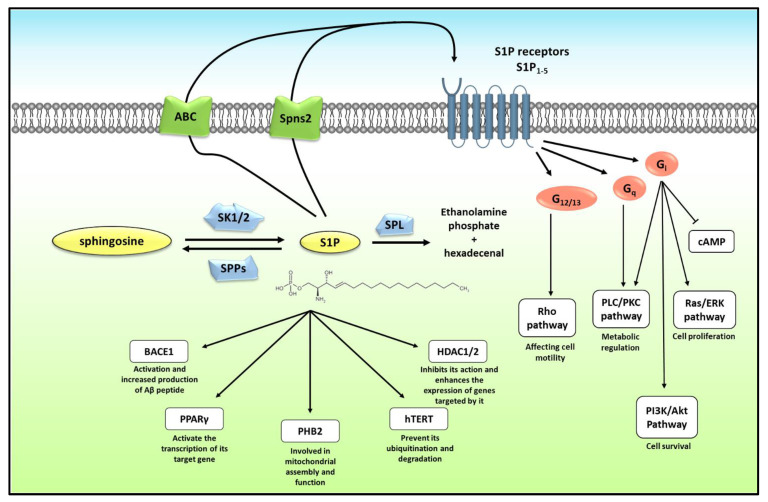
**S1P metabolism and signaling.** S1P is produced from sphingosine via SK1 or SK2 and degraded reversibly by one of the SPPs or irreversibly by SPL. Intracellularly, S1P can bind and affect the activity of different targets including BACE1, PPARγ, PHB2, hTERT, and HDAC1/2. S1P can be also secreted from the cells via ABC transporter or Spns2. Extracellular S1P binds to one of its 5 G-protein coupled receptors inducing the activation of distinct G proteins depending on the receptor, the cell type, and the state. Accordingly, extracellular S1P can activate different pathways including Rho, PLC/PKC, PI3K/Akt, and Ras/ERK pathways, thus affecting cell survival, proliferation, motility, and metabolism. S1P: sphingosine-1-phosphate, SK1/2: sphingosine kinase 1/2, SPL: S1P lyase, SPP: S1P phosphatase, BACE1: β-site amyloid precursor protein cleaving enzyme-1, PPARγ: peroxisome proliferator-activated receptor γ, PHB2: Prohibitin 2, hTERT: human telomerase reverse transcriptase, HDAC1/2: Histone Deacetylase 1/2, PLC: phospholipase C, PKC: protein kinase C, PI3K: Phosphoinositol-3-phosphate, ERK: extracellular signal-regulated kinase, S1PR: S1P receptor, Spns2: Spinster 2, cAMP: cyclic adenosine monophosphate.

**Figure 2 cancers-15-02732-f002:**
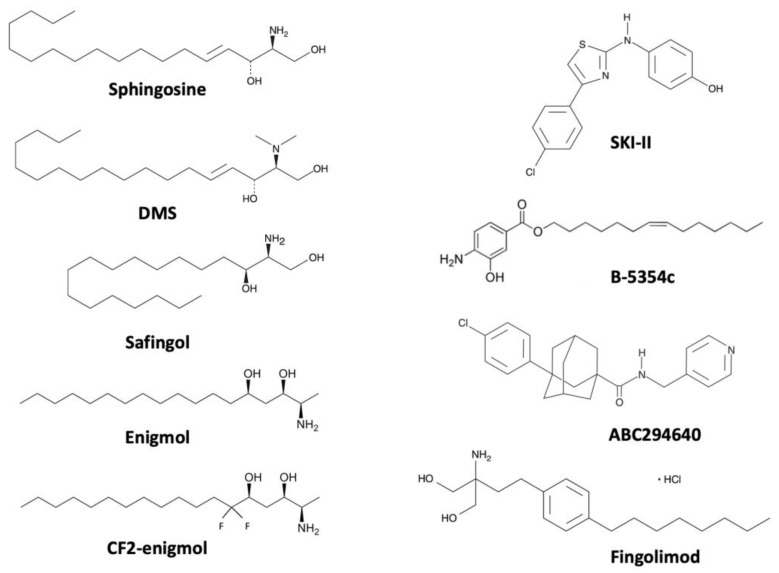
**Sphingosine and S1P metabolism-targeting molecule structures.** Comparison between sphingosine and SK inhibitors DMS, safingol, enigmol, CF2-enigmol, SKI-II, B-5354c, and ABC294640 or S1P metabolism modulator fingolimod. On the left part, molecules with the strongest structural similarities with sphingosine. On the right part, molecules with less structural similarities with sphingosine. DMS: N,N-dimethylsphingosine, Safingol: L-threo-dihydrosphingosine, Enigmol: ((2S,3S,5S)-2-amino-3,5-dihydroxyoctadecane), SKI-II: 2-(p-hydroxyanilino)-4-(p-chlorophenyl)thiazole, B-5354c: (Z)-Tetradec-7-enyl] 4-amino-3-hydroxybenzoate, ABC294640: 3-(4-chlorophenyl)-N-(4-pyridinylmethyl)-tricyclo[3.3.1.1^3,7^]decane-1-carboxamide, Fingolimod: 2-amino-2-[2-(4-octylphenyl) ethyl]propane-1,3-diol.

## Data Availability

The data can be shared up on request.

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
