# Peer review of "Targeting Sphingosine 1-Phosphate Metabolism as a Therapeutic Avenue for Prostate Cancer"

_cancers, 2023, doi:10.3390/cancers15102732_

Round 1

Reviewer 1 Report

In this review, it describes the studies demonstrating the role of S1P and S1P 29
metabolic partners in prostate cancer.
The manuscript is well-written and needs minor corrections.
1. Manuscript referencing should be corrected. For example, no reference is provided for section "1.2.2 Metastatic homing". There is no reference for lines 93 to 104. References are not considered for lines 132 to 136. Reference 23 is not related to the text.
2. The search strategy is not written.
3. There are several relevant and useful references in the context of this manuscript that is recommended to be added, for example:
doi: 10.4103/0973-1482.126446.
doi: 10.1002/1873-3468.13933
4. Manuscript's English writing needs corrections in terms of grammar.
5. It is suggested that the role of S1P in chemoresistance is written.

Manuscript's English writing needs corrections in terms of grammar.

Author Response

We thank the reviewers for their careful reading and helpful comments. As suggested by reviewer 1, to improve the readability, the language used in the entire manuscript has been revised.

Reviewer 2 Report

Overall Impression

The article titled “Targeting sphingosine 1-phosphate metabolism as a therapeutic avenue for prostate cancer” written by Saida Mebarek et al is clearly written with all the required information. Most of the sub-topics including the Role of sphingosine 1-phosphate metabolism in prostate cancer and Targeting SK/S1P metabolism as a therapeutic option for prostate cancer are well-written. Even though, some parts of the review need more clarification and corrections.

Once all these minor revisions are done, this will a good review.

Specific areas of improvement:

A.      Major corrections:  As of this, no major corrections are required. Most of the sub headings are clearly written and explored. There are no conflicts noticed and the division of the review is very accurate. Sub-topics are clearly written.

B.      Minor Corrections: Even though, the review article is well written, a few minor corrections like spelling, prepositions, and punctuations are to be verified for a perfect review. I highly recommend for preposition check and punctuation check before the final submission. Apart from some grammer corrections and small changes in prepositions and punctuations, no other minor corrections are required.

Some of the minor corrections required are listed below. Apart from that, I highly recommend making changes if necessary for the prepositions and punctuations. 

Suggestions and corrections:

1.       Prepositions should be changed in respective places.

·       Line 48, in the prostate will be correct

·       Line 60, in the bone matrix

·       Line 92, the vicious cycle

·       Line 165, product of  sphingolipid

·       Line 184, by can be removed.

·       Line 221, modulators of the S1P

·       Line 258, The first evidence

·       Line 319, related to tumoral

·       Line 352, for PC patients

2.       Corrections are needed to make sentences better

·       Line 43, after age 65 will be appropriate

·       Line 51, induce death

·       Line 65,  decreases  the quality and Worsens will be right.

·       Line 93, Tumor cells can acquire an osteoblast

·       Line 145 , A lot of therapies

·       Line 156, the discovery of

·       Line 171, turns will be the correct usage.

·       Line 174, in fact can be removed.

·       Line 181, translocates

·       Line 182, where it will be near to its substrate

·       Line 197, Final outcome can be restated as outcome as it causes redundancy.  Similiarly, line 243, or its transporter spns2. Also, line 308, long time period should be changed to long period.

·       Line 201, that confer S1P a role will be the right usage.

·       Line 208, Alzheimer’s is the correct word

·       Line 254, participate is the correct

·       Line 304, cell proliferation

·       Line 334, Oral treatment

·       Line 362, ARTAs such as

·       Line 373, local proliferation

·       Line 392, tumors expressing

·       Line 422, was efficient

·       Line 425, Sphingomab for 5 days

·       Line 435, undergo apoptosis

·       Line 436, DMS was able to induce

3.       Punctuations should be added in respective places.

·       Line 59, comma should be added after “OR”

·       Line 64, Comma should be added after example

·       Line 130, space should be removed after PC

·       Line 144, comma should be added after interest

4.       Line 360, is it SCID mice is the one one mentioned as scid? Please write the full form of scid and in bracket, you can include (SCID).

5.       Spellings should be corrected

·       Line 510 Containing

·       Line 561, Anaemia

Author Response

(The authors gave the same response as above.)

Reviewer 3 Report

In this review manuscript, Mebarek S. et. al comprehensively reviewed prostate cancer, sphingosine-1-phosphate signaling, the role of sphingosine-1-phosphate in prostate cancer development, and therapeutical inteventions against sphingosine-1-phosphate signaling for the treatement of prostate cancer. The manuscript was well-presented in a very organized manneor and and literatures were systematically reviewed. Overall, the manucripts is recommended to publish with minor editions, as recommended below. 

In section 2 line 176-179, it's highly recommended to discuss different roles of sphingosine kinase 1 and 2 in the development of prostate cancer. For instances, roles of sphingosine kinase 1 and 2 for Warburg effect has been studied in "Watson, David G., Francesca Tonelli, Manal Alossaimi, Leon Williamson, Edmond Chan, Irina Gorshkova, Evgeny Berdyshev, Robert Bittman, Nigel J. Pyne, and Susan Pyne. "The roles of sphingosine kinases 1 and 2 in regulating the Warburg effect in prostate cancer cells." Cellular signalling 25, no. 4 (2013): 1011-1017."

In section 4 and figure 2, it's highly recommended to add more inhibitor structures. For instance, reference 55 reported study of inhibitor B-5354c prostate cancer. Additionally, it's recommended to discuss selectivity of inhibitors against sphingosine kinase 1 and 2, and the resulting therapeutic outcome for prostate cancer treatment. 

The 

Author Response

We thank again the reviewers for their careful reading and helpful comments. As suggested by reviewer 1, to improve the readability, the language used in the entire manuscript has been revised.

Reviewer 4 Report

This manuscript reviews targeting S1P metabolism as strategy for prostate cancer. The story started to overview the prostate cancer and the SIP signaling pathway, respectively, and followed by establishing the relationship between prostate cancer and S1P, in the end several examples of active molecules were cited. Overall, this review serves a good tutorial for the researches who were involved in the field of targeting S1P. Therefore, as far as I am concerned, this manuscript can be assigned as minor revision.

Here are the questions and recommendations.

1. In section 4, the compounds listed are pretty famous compounds, most of which has been used or investigated for more than 10 years now. For the reviews in 2023, it might be good to add some new compounds which are published in the recent years, this may include some new version of known compounds, such as SKI-V (doi: 10.7150/ijbs.71381), SK1/SK2 dual inhibitor, some structures with improved functionality (doi: 10.1016/j.bmc.2020.115941), etc.

2. The other question is about the potential problem of targeting S1P. In the past decades, are there any challenges mentioned in the literature? Such as off target effect? The inhibitor becomes the activator?

Author Response

(The authors gave the same response as above.)
